# Indices of Informational Association and Analysis of Complex Socio-Economic Systems

**DOI:** 10.3390/e21040367

**Published:** 2019-04-04

**Authors:** Paulo L. dos Santos, Noé Wiener

**Affiliations:** 1Department of Economics, New School for Social Research, New York, NY 10003, USA; 2Department of Economics, University of Massachusets, Amherst, NY 01002, USA

**Keywords:** information theory, complex socio-economic systems, mathematical economics, mathematical sociology

## Abstract

This paper is motivated by a distinctive appreciation of the difficulties posed by quantitative observational inquiry into complex social and economic systems. It develops ordinary and piecewise indices of joint and incremental informational association that enable robust approaches to a common problem in social inquiry: grappling with associations between a quantity of interest and two distinct sets of co-variates taking values over large numbers of individuals. The distinct analytical usefulness of these indices is illustrated with their application to inquiry into the systemic economic effects of patterns of discrimination by social identity in the U.S. economy.

## 1. Introduction

This paper draws on information theory and political economy to make a methodological and instrumental contribution to observational inquiry into the functioning of economic and broader social systems. It is most broadly motivated by a recognition of the those systems are *complex*. Complexity creates formidable and largely ignored conceptual and practical difficulties for analyses grounded on the strong form of individualist reductionism dominating current economic thought and informing most quantitative observational work in economics.

The paper discusses how a wide and plural range of contributions to economic thought have effectively advanced an alternative methodological foundation for economic analysis. This alternative is based on the recognition that the observable empirical regularities upon which we may ground our understanding of the functioning of economic systems are generally systemic and independent of much of the fine-grained detail of individual behavior, knowledge, and interactions. The paper looks to information theory to develop observational tools enabling characterizations of those systemic regularities and of the reduced-form associations they define between observable individual characteristics. The concepts of *entropy* and *mutual information* offer very general, non-parametric, informational measures of those associations: How much of our ignorance or lack of knowledge about individual values of a given variable is removed once we observe individual values of other variables. They offer innovative bases for observational inquiry into complex economic systems.

The paper derives a series of ordinary and piece-wise indices of informational association that enable robust observational approaches to a common problem in inquiry into complex economic and social systems: grappling with the systemic associations between a variable of interest $X_{0}$ and a set of generally interdependent covariates $X_{k}$ that may be meaningfully decomposed into two distinct and exhaustive subsets $X_{i}$ and $X_{e}$, taking values over large numbers of individuals. Those indices make it possible to develop non-parametric measures of the extent to which a set $X_{k}$ informationally accounts for a quantity $X_{0}$; the informational synergies and redundancies between $X_{i}$ and $X_{e}$ in those accounts; and of the extent to which a subset $X_{i}$ has an informational association with $X_{0}$ independently of the quantities in $X_{e}$ (and *vice versa*). They can thus guide observational inquiry and provide observational foundations for systemic theorizations of the functioning of economic systems.

To illustrate the distinctive usefulness of the indices it develops, the paper briefly reports on their recent application to analysis of the observable economic effects of patterns of discrimination by social identity on distributions of individual income in the U.S. economy [1]. This application is shown to improve on extant statistical tests for the presence of economic effects of discrimination; elicits a discussion on how the indices the paper develops relate and contribute to the use of entropy-based measures of income inequality [2]; and results in innovative instruments and diagnostics for the presence of “equality of opportunity,” as understood by political and economic philosophers [3,4,5,6]. Finally, it also yields distinctive insights into the manner in which observed distributions of income in the U.S. embody forms of discrimination, and into the iniquitous manner in which levels of educational attainment influence incomes across different social-identity groups.

The paper is organized as follows. Section two offers the central methodological discussion motivating and informing the paper. Section three explains the content and usefulness to analysis of complex economic systems of the concepts of entropy and mutual information. Sections four and five formally develop the ordinary and piecewise indices of joint, incremental, and mutual information advanced by the paper. Section six contains the illustrative application of these indices to analysis of the effects of social identity on individual incomes. A brief concluding section closes the discussion.

## 2. Individuals, Complexity, and Observable Outcomes

Information-theoretic measures can offer robust, non-parametric characterizations of reduced-form, systemic associations between observable quantities generated by decentralized, market economies. Those characterizations are useful to inquiry into complex relationships and effects that generally defy analysis based on the strong individualist reductionism that dominates contemporary economic thought. They require no detailed descriptions of agent characteristics, specific market equilibria, or particular mechanisms. However, understanding the results they yield requires a distinctive, social understanding of the interplay between individual agencies and systemic interdependencies in conditioning observable economic outcomes, including the emergence of systemic regularities that may inform the development of theory. This requires stepping back from the accepted methodological approach informing much of today’s economic analysis.

### 2.1. Neoclassical Individualist Reductionism

Contemporary economics has settled on a rather strong form of individualist reductionism. The substance and disciplinary dominance of this methodological stance was summarized well by leading neoclassical theorist Kenneth Arrow,
“It is a touchstone of accepted economics that all explanations must run in terms of the actions and reactions of individuals. Our behavior in judging economic research, in peer review of papers and research, and in promotions, includes the criterion that in principle the behavior we explain and the policies we propose are explicable in terms of individuals, not of other social categories.”[7]

In line with this proposition, contemporary economics has sought to characterize the functioning of economic systems on the basis of detailed descriptions of the behavior of “representative” individuals, or of similarly detailed game-theoretic representations of interactions between small numbers of agent types. In canonical microeconomic and macroeconomic frameworks, this includes specifications of subjective consumption preferences; technological constraints; knowledge states; the specific forms in which individuals define and pursue their self-interest; and the institutional forms conditioning competitive market interactions; and the deterministic equilibrium states they define [8,9]. While possibly useful as bases for pursuing thought exercises, these approaches face practical and conceptual difficulties that are not widely acknowledged among economists.

The conceptual difficulties with the idea of seeking to understand economic systems on the basis of an abstraction founded on a would-be “representative” individual have been well understood among dissenting economists [10]. More general attempts to interpret what we can observe in economic systems in relation to strongly specified descriptions of the intentions and actions of any number of individuals also face great challenges.

The only setting in which the data we observe conceivably reflects the intentions of all individuals is one in which our measurements are consistently taken from an economy in a state of deterministic equilibrium in every market, at which all individual plans are not only formulated on the basis of things we can observe, but are also being successfully implemented. However, we cannot expect to observe real-world economies in such states of general equilibrium—both because adjustments to such a state take time, and because the factors conditioning the characteristics of any would-be general equilibrium are overwhelmingly likely to be changing faster than any such adjustment. We should expect typically to observe individual outcomes at variance with individual plans [11].

More broadly, the individual characteristics and behaviors upon which individualist theorizations are predicated are often unobservable and possibly unintelligible to observers [12], straining the scientific soundness of suppositions made about their nature [13]. Even if it were somehow possible to overcome these obstacles to develop observationally well-founded characterizations of individual behavior, those would still not generally offer a robust basis to understand the macroscopic functioning of economic systems. As well established across a variety of other disciplines, detailed descriptions of individual behavior are generally impractical bases to investigate the functioning of large, complex systems composed of many non-linearly coupled parts. Individual evolutions in such systems typically trace chaotic, disequilibrium paths along which it is very difficult to relate what we can observe to posited details of individual behavior.

Economic and social systems pose an additional and characteristic difficulty relative to physical systems in this regard: all economically relevant features of individuals are themselves shaped by economic competition and broader social interactions. It is not just that the rules of the economic game are social, as Kenneth Arrow pointed out. The players themselves are social too: Individual consumer preferences are shaped by the competitive interventions of enterprises [14]; economies of scale, scope, and agglomeration ensure the productive capacities of firms reflect their history of competitive interactions and geographical contexts [15]; and the manners in which individuals define and choose to pursue their self-interest are notoriously shaped by a variety of social influences, including fads, “herd behavior,” social power, etc. [16,17].

If the characteristics of economic individuals evolve as part of competitive market interactions, taking them as an analytical starting point is not just impractical. It is an arbitrary *parametrization* of the complex, dynamic functioning of a competitive, decentralized market economy. Arbitrary because there should be no *a priori* reason to expect regularities or equilibria allowing us to characterize the functioning of such economies to be defined by stability in micro-level details of individual characteristics. Individualist parametrizations also have a conservative thrust, in the sense that they focus analytical attention on the consequences of given differences in economic characteristics across individuals, regions, or economies, with less emphasis on what are often the most interesting and pressing questions of economic inquiry: the social and historical processes giving rise to those present differences. This practice has a long pedigree in economic analysis, going back at least as far as David Ricardo’s discussion of comparative advantage. That account motivates the benefits of trade on the basis of a thought exercise where Portugal has a given comparative advantage in the production of wine—a good whose production is most strongly influenced by inherent climatological characteristics of a region—while England has a comparative advantage in the production of cloth—a good reflecting a history of industrialization that paves the way for subsequent gains in labor productivity.

Finally, the annual or quarterly frequencies at which we are typically able to observe some elements of individual economic states are far lower than the frequencies at which individuals interact. The quantities we can observe reflect not the behavior of individuals *per se*, but the accumulated, reduced-form result of many interactions among large numbers of individuals. Between the times at which we can take measurements, much of the information about the detail of individual behavior has been lost—both as a result of large numbers of interactions, and of changes to the individuals themselves. It is generally impossible in those cases to draw on observation to inform the kinds of detailed, reductionist accounts sought by most economists.

Developing an observationally grounded understanding of the functioning of contemporary economic systems requires grappling deliberately with these difficulties.

### 2.2. The Systemic, Social Content of What We Observe

Contributions from a diverse range of traditions in economics and political economy have shown that reductions of observable economic outcomes to detailed descriptions of individual states and actions are unnecessary and often misleading. Many of them have also effectively contended that the observable regularities that enable economists to offer characterizations of the functioning of decentralized, market economies are *systemic* and *social*. Those regularities are not generally defined by the fine-grained detail of individual intentions and actions or by the equillibra those agencies may condition. They reflect emergent outcomes of competitive interactions and interdependencies that duly systemic theorizations of the social outcomes of market interactions may explain.

Observable economic outcomes may be robustly indifferent to much of the micro-level detail of individual behavior [18]. Basic postulates of economic analysis can be explained as results of rather simple propositions, without recourse to detailed descriptions of that behavior. The “law of demand” can be understood to reflect simple budgetary constraints bearing on all consumers, and not as a consequence of their “rational” optimizing behavior and of the transitivity, completeness, and reflexivity of their preferences across available bundles of goods [19]. The observation that factor shares in the output of enterprises typically add up to one can be understood to follow from similar accounting facts, and not as validation of the theory of “perfect competition” and of the presence of technological constraints generating constant-returns-to-scale, Cobb-Douglas production functions [20].

Many important economic outcomes can also be understood as emergent results of competitive interactions and structural interdependencies, and irreducible to the details of the intentions, knowledge, and actions of any individual. This was first recognized by Adam Smith [21], who famously noted that competition can ensure that the pursuit of pecuniary self interest unintentionally gives rise to coordination across large numbers of economic individuals, and to a socially desirable push toward improvements in the physical productivity of labor and in the social measure of prosperity. There is considerable irony in the dominance of individualist reductionism in contemporary economics. The early development of the discipline was motivated by the gradual recognition that societies increasingly dependent on market interactions were subjected to influences beyond the control of individuals and traditional institutions. It is thus no coincidence that the central, towering founding figure of the discipline may have also been the first scholar in any field to write about what we now term *emergence*.

Friedrich Hayek offered a more contemporary, radical-subjectivist elaboration of this view [22]. The knowledge necessary to achieve coordination and desirable social outcomes is dispersed, subjective knowledge of localized conditions and profit opportunities. It cannot be acquired or even understood by any single agent. It is competition between alert agents that ensures prices spontaneously come to reflect the comparative social significance of all opportunities. The Efficient Markets Hypothesis can be understood as an influential version of this view, applied to the content and evolution of observable capital-market prices [23]. In both cases, market competition is understood to give rise to observable price systems offering emergent, social quantifications of a broad range of detailed economic realities that no single agent may observe or characterize, including theorists of economic functioning. In both cases, economic analysis actually results in strong statements concerning our *ignorance* about those details and their future effects on prices [24].

Other traditions of economic thought also have emphasized how structural interdependences and competitive interactions ensure that the observable regularities upon which we may base our understanding of the functioning of economic systems reflect non-trivial, unintended, and at times perverse effects of individual actions. Keynesian contributions have pointed to the evolution of individual balance-sheet positions, which is shaped by the aggregate identity between incomes and expenditures of all participants in an economy. The resulting interdependencies can condition paradoxical results or fallacies of composition, such as the *paradox of thrift* and the *paradox of debt* [25,26].

Karl Marx identified another potential source for paradoxes and fallacies of composition in the competitive pursuit of technical innovation: the dependence of market prices on *average* measures of physical productivity across all suppliers in an industry [27]. A recent contribution has offered a generalized version of this argument, suggesting that the observed outcomes of many complex competitive interactions reflect a simple, emergent outcome: the *social scaling* of certain individual characteristics by social or average measures of themselves across all competing individuals [28].

These and related contributions suggest that the things we can typically measure in economic analysis are reduced-form outcomes of the complex, dynamic interplay between evolving individual agents, competitive interactions, and structural interdependencies. Any regularities present in them are most usefully understood as emergent, systemic or social outcomes of those processes. They require and enable the formulation of systemic or social characterizations of the outcomes of competitive functioning in decentralized, market economies. The development of those characterizations generally requires careful consideration of what elements of individual agency, competitive interactions, and structural interdependencies prove relevant to the determination of what we observe. There is no justification for a prior commitment to theoretical accounts privileging only one of these interrelated features of economic interactions as a methodological foundation.

The information theoretic indices and broader methodological approach offered below can help the development of such characterizations. By so doing, they may also support the development of new, observationally grounded political economies of the social content of contemporary capitalist economies.

## 3. Drawing on Information Theory

Information Theory offers distinctively useful tools enabling formal inferences about complex patterns of economic and social interaction based on their observable outcomes. This section discusses two central information theoretic concepts that can help guide observational inquiry into the associations between economic characteristics across large numbers of individuals: entropy and mutual information.

### 3.1. The Formal Setting

To motivate their applicability for analysis of social systems, consider an economic or social system as composed of a large number *N* of individual members. At any given point in time, each of those members has an individual state defined over a set of *v* degrees of freedom, $X_{v}=\left\{X_{0},X_{1},\ldots ,X_{v}\right\}$. Individual degrees of freedom may describe quantifiable individual characteristics, as well as macroscopic quantities that take the same value across a large number of individuals in the system. They may also describe qualitative or categorical individual characteristics, including descriptions of an individual’s institutional or relational situations. Coding schemes mapping the latter characteristics onto distinct real numbers allow individual states to be represented by vectors ${\vec{x}}_{v}=\left\{x_{0},x_{1},\ldots ,x_{v}\right\}$, with the set of all such individual states denoted by $T\subseteq R^{v}$.

In all practical work the space *T* is effectively “coarse grained” into $s_{v}$ bins or effective individual states. The *phase space*
$\Omega_{v}$ of this system can then be thought of as the set of all its possible micro-level configurations—all the $s_{v}^{N}$ arrangements of individual members of the system across the $s_{v}$ individual states. Since $X_{v}$ is an exhaustive description of all individual characteristics relevant to economic and social interactions, the functioning of the system is entirely indifferent between individuals with the same ${\vec{x}}_{v}$. What will matter is the total number of members or occupancy $n_{i}$ in each of the $s_{v}$ bins in *T*. The macroscopic state of the system can be defined as a frequency function $f\left({\vec{x}}_{v}\right)$ describing the normalized occupancy of each bin. The functioning of a system defines an $\left(s_{v}-1\right)$-dimensional space $\Phi_{v}$ containing all macroscopic states $f\left({\vec{x}}_{v}\right)$ the system may occupy. The laws and regularities that define a system are given statistical expression in the shape of $\Phi_{v}$.

In observational, quantitative social inquiry we often face variations of the following analytical problem within this setting: We can observe the values taken by w<v individual degrees of freedom across n<N members of the system. This allows construction of frequency histograms $f\left({\vec{x}}_{w}\right)$ over the values taken by the vector ${\vec{x}}_{w}$ of observed individual states. We have limited knowledge of the micro-level interactions driving the functioning of the system. In addition, we generally do not know the full set $X_{v}$ of relevant degrees of freedom. However, we would like to draw on what we observe to infer as much as we can about the functioning of the social or economic system at hand. Formally, we want to develop increasingly accurate descriptions of the shape of $\Phi_{w}\subset \Phi_{v}$.

### 3.2. Entropy and Mutual Information

In this connection the concept of *entropy* is distinctively useful. The entropy $H\left(X\right)$ for any set of degrees of freedom X in a system occupying a macroscopic state $f\left(\vec{x}\right)$, defined over *s* bins, can be understood as an average, logarithmic measure of the number $W_{f}$ of micro-level configurations or elements in the phase space Ω yielding the macroscopic state $f\left(\vec{x}\right)\in \Phi$. Formally,
(1)$$H\left(X\right)=\frac{\log W_{f}}{N}=\frac{1}{N}\log\frac{N!}{\prod_{i}^{s}n_{i}!}\approx -\sum_{i=1}^{s}f_{i}\log f_{i}$$

Note that this quantity can also be understood as a measure of the diversity or heterogeneity in the values taken by X. If all individuals are in the same bin, Equation (Equation 1) ensures entropy is zero. If individuals are evenly distributed across all *s* bins—a state of maximum diversity or heterogeneity—entropy reaches its maximum value: logs. It should be obvious that a change in the state of a single individual results in an increase in entropy if and only if the change takes that individual to a state with lower occupancy than the state it originally occupied. That is, entropy increases only when diversity or heterogeneity increases [29].

Entropy is useful in analysis of systems with large N>>m for at least two reasons. First, for those systems the combinatorial dominance of the distribution $f^{*}\left(\vec{x}\right)$ achieving maximum entropy over all other macroscopic states in Φ is overwhelming. This conclusion can guide the iterative process of observational inquiry into the functioning of such systems. If we have a set of knowledge, beliefs, or hypotheses *G* suggesting that the functioning of the system keeps it within a set $\Phi^{G}$ of macroscopic states, we should expect to observe macroscopic behavior in line with the state $f^{*}\left(\vec{x}|G\right)$ that maximizes entropy over that set. This is the *Principle of Maximum Entropy* (PME).

It is important to note that the PME is not a behavioral hypothesis and is entirely independent of the elements in set *G*. In fact, if we observe macroscopic behavior at variance with $f^{*}\left(\vec{x}|G\right)$, the PME tells us that *G* is either incomplete or wrong, informing subsequent inquiry [30]. What the Principle offers is a distinctive and logically robust way to link knowledge we may have about the micro-level functioning of a system and what basic combinatorial considerations lead us to conclude about its observable macroscopic states. This is a very different conceptualization of the relationship between micro- and macro-level functioning than that which grounds most contemporary economic thinking.

A converse application of the PME is particularly useful in observational work in quantitative social inquiry [29,31,32,33,34,35,36]. Sometimes we observe cross-sectional frequencies $f\left({\vec{x}}_{w}\right)$ that are persistently and ubiquitously well described by known, closed-form functional forms. Those functional forms are often entropy maxima over sets $\Phi_{w}$ that are defined by known moment constraints on the distribution of ${\vec{x}}_{w}$.

We may infer that those moment constraints offer good systemic descriptions of laws or regularities present in the processes conditioning values of ${\vec{x}}_{w}$. Interactions involving all observed and non-observed degrees of freedom in $X_{v}$ yield outcomes in $\Phi_{w}$ that are aptly characterized by them. Those constraints often provide important formal clues about the macroscopic or *social* content of the micro-processes at hand, and can inform the development of empirically successful economic or social theories [28,37,38].

Entropy is also useful in the more common settings where observed distributions are not well described by known, closed-form functional forms. Entropy can be understood as a measure of the uncertainty we have about the exact micro-level configuration of a system we observe at a given macroscopic state. Depending on the base of the logarithm used in definition (Equation 1), entropy measures the average number of bits, nats, or dits necessary to enumerate all micro-level configurations resulting in that state.

This measure of uncertainty motivates the concept of *mutual information*. For two degrees of freedom $X_{i}$ and $X_{j}$, it is defined by,
(2)$$I\left(X_{i},X_{j}\right)=H\left(X_{i}\right)-H\left(X_{i}|X_{j}\right)=H\left(X_{i}\right)-\sum_{x_{j}}f\left(x_{j}\right)H\left(X_{i}|x_{j}\right)$$

This is a quantification of the average reduction in our uncertainty about $X_{i}$ when we observe the distribution of $X_{j}$: The change in the average number of bits, nats, or dits needed to enumerate or identify uniquely each micro-level configuration compatible with observation when moving from observing only $X_{i}$ to observing $X_{i}$ and $X_{j}$. Mutual information can also be thought of as a measure of the information shared between the two quantities, in that it quantifies how much we learn about one of them from observation of the other.

## 4. Indices of Informational Association

The concepts of entropy and mutual information enable the development of non-parametric characterizations of informational associations present in observed data generated by economic and broader social interactions. Those can inform the development of systemic characterizations of the functioning of economic systems. In line with that objective, this section develops a series of indices of mutual, joint, and incremental or conditional mutual information between sets of individual degrees of freedom. It derives versions of these indices that are useful for a common situation in social inquiry: A setting where we are interested in the comparative influence of two sets of factors over a variable of interest.

### 4.1. Multivariate Mutual and Joint Information

The multivariate generalization of mutual information requires careful consideration. As motivated by canonical contributions to information theory, [39,40,41], note that $I\left(X_{i},X_{j}\right)=I\left(X_{i}\right)-I\left(X_{i}|X_{j}\right)$, where the self mutual information $I\left(Y\right)=H\left(Y\right)$. By extension,
(3)$$I\left(X_{0},X_{1},X_{2}\right)=I\left(X_{0},X_{1}\right)-I\left(X_{0},X_{1}|X_{2}\right)$$
where the *conditional or incremental mutual information*
$I\left(X_{0},X_{1}|X_{2}\right)=H\left(X_{0}|X_{2}\right)-H\left(X_{0}|X_{1},X_{2}\right)$ measures the information gained about $X_{0}$ upon observation of $X_{1}$ when $X_{2}$ is already known. The tripartite mutual information in (Equation 3) is a measure of the information shared by all three variables: the information shared by $X_{0}$ and $X_{1}$, minus the part of that shared information not contained in $X_{2}$.

The general multivariate mutual information can be defined recursively,
(4)$$I\left(X_{0},X_{1},\ldots ,X_{k}\right)=I\left(X_{0},X_{1},\ldots ,X_{k-1}\right)-I\left(X_{0},X_{1},\ldots ,X_{k-1}|X_{k}\right)$$

The mutual information between all k+1 variables measures the shared informational content of the first *k* variables minus the part of that content not contained in $X_{k}$.

In inquiry into the functioning of economic and social systems, a different measure of informational association is more directly and obviously useful. We are often interested in learning not about the informational content shared among several variables but in how much of the uncertainty in a single degree of freedom $X_{0}$ is removed when we observe values taken by a set $X_{k}=\left\{X_{1},\ldots ,X_{k}\right\}$ of other degrees of freedom. Put differently, we are often interested in the *informational account* of $X_{0}$ given jointly by the elements in $X_{k}$: How much do we know about individual values $x_{0}$ taken by $X_{0}$ based on knowledge or observation of $\chi_{k}=\left\{x_{1},\ldots ,x_{k}\right\}$.

This may come up as part of general inquiry into the dynamic co-determinations between all these variables. It may also come up in settings where we know that the elements of a set $X_{k}$ are prior to the interactions generating values of all other degrees of freedom in a system, including $X_{0}$. In such cases, the informational equivalence may be taken as a measure of the extent to which the elements in $X_{k}$ influence values of $X_{0}$, directly or indirectly through their influence on other degrees of freedom.

To characterize this kind of informational accounting, a measure of *joint mutual information* is more useful [42,43,44]. Defining it first for a setting with three degrees of freedom, consider,
(5)$$I\left(X_{0};\left(X_{1},X_{2}\right)\right)=H\left(X_{0}\right)-H\left(X_{0}|\left(X_{1},X_{2}\right)\right)$$

Which measures the reduction in uncertainty about values of $X_{0}$ once values of $X_{1}$ and $X_{2}$ are taken into account. The relationship between this measure and the conditional mutual information can be easily established. Adding $H\left(X_{0}|X_{1}\right)-H\left(X_{0}|X_{1}\right)=0$ to this definitions yields,
(6)$$\begin{array}{c} I\left(X_{0};\left(X_{1},X_{2}\right)\right)=\left(H\left(X_{0}\right)-H\left(X_{0}|X_{1}\right)\right)+\left(H\left(X_{0}|X_{1}\right)-H\left(X_{0}|\left(X_{1},X_{2}\right)\right)\right)\\ =I\left(X_{0},X_{1}\right)+I\left(X_{0},X_{2}|X_{1}\right) \end{array}$$

The joint mutual information between $X_{0}$ and $\left(X_{1},X_{2}\right)$ is the sum of the mutual information between $X_{0}$ and $X_{1}$ and a conditional or incremental mutual information—the information gained about $X_{0}$ upon observation of $X_{2}$ when $X_{1}$ is already known. This results in a measure of the total reduction in uncertainty about $X_{0}$ arising from joint observation of $X_{1}$ and $X_{2}$.

The multivariate generalization of this measure for $X_{0}$ and a set $X_{k}$ of *k* other degrees of freedom that may take individual values $\chi_{k}=\left\{x_{1},x_{2},\ldots ,x_{k}\right\}$ may also be defined recursively,
(7)$$\begin{array}{c} I\left(X_{0},X_{k}\right)=H\left(X_{0}\right)-H\left(X_{0}|X_{k}\right)\\ =\left(H\left(X_{0}\right)-H\left(X_{0}|X_{k-1}\right)\right)+\left(H\left(X_{0}|X_{k-1}\right)-H\left(X_{0}|X_{k}\right)\right)\\ =I\left(X_{0},X_{k-1}\right)+I\left(X_{0},X_{k}|X_{k-1}\right) \end{array}$$

### 4.2. A Useful Decomposition

While it is possible in principle to characterize measures of joint, mutual, and incremental informational association between all possible groupings of elements in $X_{k}$ and $X_{0}$, the resulting decompositions are impractically cumbersome even for small *k*. Fortunately, in many applications in economic and social inquiry, we are interested in a simpler decomposition: separating the variables in $X_{k}$ into two mutually exclusive sets, $X_{e}$ containing *e* of the *k* individual degrees of freedom, and its complement in $X_{k}$, $X_{i}$, containing the remaining i=k−e ones,
(8)$$\begin{array}{c} I\left(X_{0};X_{k}\right)=\left(H\left(X_{0}\right)-H\left(X_{0}|X_{e}\right)\right)+\left(H\left(X_{0}|X_{e}\right)-H\left(X_{0}|X_{k}\right)\right)\\ =I\left(X_{0};X_{e}\right)+I\left(X_{0};X_{i}|X_{e}\right)\\ =I\left(X_{0};X_{i}\right)+I\left(X_{0};X_{e}|X_{i}\right) \end{array}$$

The total, joint informational association of the degrees of freedom in $X_{k}$ and the variable of interest $X_{0}$ is given by the joint mutual information between the latter and the variables in the set $X_{e}$ plus the incremental information gained about $X_{0}$ upon observation of $X_{i}$ when $X_{e}$ is already known. The symmetric decomposition is also valid, naturally.

Since the degrees of freedom in each two sets $X_{e}$ and $X_{i}$ are being considered jointly, it is also possible to consider the tripartite mutual information,
(9)$$I\left(X_{0},X_{e},X_{i}\right)=I\left(X_{0},X_{e}\right)-I\left(X_{0},X_{e}\right|X_{i})=I\left(X_{0},X_{e}\right)+I\left(X_{0},X_{i}\right)-I\left(X_{0},X_{k}\right)$$

The joint mutual information in (Equation 8) can be decomposed into the two measures of conditional or incremental mutual information defined by $X_{e}$ and $X_{i}$ and the mutual information between $X_{0}$ and the two sets.
(10)$$I\left(X_{0};X_{k}\right)=I\left(X_{0};X_{e}|X_{i}\right)+I\left(X_{0};X_{i}|X_{e}\right)+I\left(X_{0},X_{e},X_{i}\right)$$

The expressions of decompositions (Equation 8) and (Equation 10) as normalized indices of informational association enable the pursuit of an innovative, systemic approach to observational, quantitative social inquiry.

### 4.3. Coefficients of Association and Informational Accounts

The joint mutual information between $X_{0}$ and $X_{k}$ measures the extent to which the former is informationally equivalent to the latter. We may thus consider the *informational account* of $X_{0}$ provided by $X_{k}$. The measure of this joint mutual information normalized by the entropy of $X_{0}$ offers a useful measure of the success of this informational account,
(11)$$A\left(X_{0}\|X_{k}\right)\equiv \frac{I\left(X_{0},X_{k}\right)}{H\left(X_{0}\right)}=1-\frac{H\left(X_{0}|X_{k}\right)}{H\left(X_{0}\right)}$$

It should be obvious that $A\left(X_{0}\|X_{k}\right)\in \left[0,1\right]$, with $A\left(X_{0}\|X_{k}\right)=1$ only when the account is *deterministic*: There is no information about $X_{0}$ outside of the set $X_{k}$. If we know all values $\chi_{k}$ taken by $X_{k}$, we have exact knowledge of the individual values taken by $x_{0}$ by all individuals in the system. We may term a degree of freedom $X_{i}$ in an account provided by $X_{k}$
*independent* if $A\left(X_{i}\|X_{j}\right)=0,\;\forall X_{j}\in X_{k},j\neq i$. An account may be termed *orthogonal* if all the degrees of freedom involved are independent.

There should be no expectation that analysis of complex social systems can even approximately result in accounts that are deterministic or orthogonal. The general expectation of orthogonality in accounts of social phenomena is misguided and partly conditioned by confusion between parametrizations as a thought exercise, and the practical possibility of exerting control over dynamically interdependent quantities in complex social systems. There are simply too many interdependences and informational interactions between the degrees of freedom involved at observable frequencies. However, in social inquiry, we can often make some progress toward understanding the influences on a degree of freedom $X_{0}$ by considering measures of its incremental and mutual informational association with two mutually exclusive subsets of $X_{k}$, $X_{e}$ and $X_{i}$,
(12)$$I_{X_{e}|X_{i}}\equiv \frac{I\left(X_{0};X_{e}|X_{i}\right)}{H\left(X_{0}\right)};\;\;\;I_{X_{i}|X_{e}}\equiv \frac{I\left(X_{0};X_{i}|X_{e}\right)}{H\left(X_{0}\right)};\;\;\;M\left(X_{0},X_{e},X_{i}\right)\equiv \frac{I\left(X_{0},X_{e},X_{i}\right)}{H\left(X_{0}\right)}$$

These conventions permit several different ways to express the decomposition of $A\left(X_{0}\|X_{k}\right)$,
(13)$$\begin{array}{c} A\left(X_{0}\|X_{k}\right)=A\left(X_{0}\|X_{e}\right)+\left(1-A\left(X_{0}\|X_{e}\right)\right)A\left(X_{0}|X_{e}\|X_{i}\right)\\ =A\left(X_{0}\|X_{e}\right)+I_{X_{i}|X_{e}}=A\left(X_{0}\|X_{i}\right)+I_{X_{e}|X_{i}}\\ =I_{X_{i}|X_{e}}+I_{X_{e}|X_{i}}+M\left(X_{0},X_{e},X_{i}\right) \end{array}$$

The total proportional reduction in uncertainty about $X_{0}$ can be divided into the coefficient of unconditional informational association between $X_{0}$ and one of the two sets of individual degrees of freedom, and the coefficient of incremental informational association between $X_{0}$ and the other set of individual degrees of freedom. It can also be expressed as a sum of the two coefficients of incremental informational association, minus the coefficient of mutual informational association between $X_{0}$, $X_{e}$, and $X_{i}$.

The magnitude and sign of $M\left(X_{0},X_{e},X_{i}\right)$ reveal an important informational relationship between these degrees of freedom. They quantify the *redundancy* or *synergies* in the informational association of $X_{0}$ with $X_{e}$ and $X_{i}$.

Formally, the sets of degrees of freedom $X_{e}$ and $X_{i}$ exhibit a measure of redundancy in an informational account of $X_{0}$ when $M\left(X_{0},X_{e},X_{i}\right)>0$, which requires that $A\left(X_{0}\|X_{k}\right)>I_{X_{i}|X_{e}}+I_{X_{e}|X_{i}}$. There is information about $X_{0}$ in $X_{e}$ that is also shared by $X_{i}$, and *vice versa*.

The less immediately intuitive setting of informational synergies occurs when $M\left(X_{0},X_{e},X_{i}\right)<0$, which requires that $I_{X_{i}|X_{e}}>A\left(X_{0}\|X_{i}\right)$, which also implies that $I_{X_{e}|X_{i}}>A\left(X_{0}\|X_{e}\right)$. That is, there are informational synergies between the two sets of degrees of freedom in an account of $X_{0}$ whenever the incremental information of each set of degrees of freedom $X_{e}$, $X_{i}$ with $X_{0}$ is greater than its respective unconditional informational association with $X_{0}$. In those cases, knowledge of one set of degrees of freedom reduces more uncertainty about $X_{0}$ if the other degree of freedom is already known. There is information about $X_{0}$ in the *combination* of $X_{e}$ and $X_{i}$ that is not contained in either of those two sets individually. The combination may be associated with further degrees of freedom associated with $X_{0}$.

## 5. Categorical Characteristics and Piecewise Decompositions

In analysis of economic and social systems we are often confronted with categorical individual characteristics. In those cases, part-piecewise decompositions of the indices developed above can be useful in inquiry into the emergent, social consequences of certain individual characteristics on specific individual outcomes. This section develops such decompositions for coefficients of joint informational association and for coefficients of mutual and incremental associations involving a variable of interest and two sets of individual degrees of freedom.

### 5.1. Piecewise Joint Associations

Let $X_{k}$ be a set of observable categorical individual degrees of freedom, divided as above into two subsets $X_{e}$ and $X_{i}$. Let $X_{0}$ be a quantitative variable of interest. In this setting, it is possible to derive the part-pointwise decomposition of the coefficient of informational association defined in (Equation 13) across all individual values $\chi_{k}$ taken by $X_{k}$,
(14)$$A\left(X_{0}\|X_{k}\right)=\sum_{\chi_{k}}f\left(\chi_{k}\right)a\left(Y\|\chi_{k}\right);a\left(X_{0}\|\chi_{k}\right)\equiv \frac{H\left(X_{0}\right)-H\left(X_{0}|\chi_{k}\right)}{H\left(X_{0}\right)}$$

The coefficients of joint pointwise informational association $a\left(X_{0}\|\chi_{k}\right)$ measure the proportional reduction in heterogeneity or observer uncertainty about $X_{0}$ once it is verified that $X_{k}=\chi_{k}$.

It is important to note that (Equation 14) ensures that the overall informational association of a set of degrees of freedom $X_{k}$ and $X_{0}$ may be due to very different informational associations between $X_{0}$ and sub-populations $\chi_{k}$. In fact, while $A\left(X_{0}\|X_{k}\right)$ is always non-negative, values of $a\left(X_{0}\|\chi_{k}\right)$ may be negative. This occurs when the distribution of $X_{0}$ across all members of a $\chi_{k}$ sub-population has greater heterogeneity than the distribution across the population as a whole. In those cases, the measures in $X_{k}$ have a greater informational influence on $X_{0}$ within sub-populations with $X_{k}\neq \chi_{k}$ than on incomes for individuals with $\chi_{k}$ characteristics. Put differently, factors *other than those contained in or associated with*
$X_{k}$ have a greater informational role in shaping the heterogeneity of $X_{0}$ within sub-population $\chi_{k}$ than within the population as a whole.

In general, observed differences in measures of $a\left(X_{0}\|\chi_{k}\right)$ across different $\chi_{k}$ sub-populations are very significant in large-*N* systems. They strongly suggest that the processes determining individual values of $X_{0}$ across those sub-populations are formally different.

To see this, consider two such sub-populations $\chi_{k}=\alpha ,\beta$, with $N_{\alpha }$ and $N_{\beta }$ members, and observed piecewise coefficients of association $a\left(X_{0}\|\beta \right)-a\left(X_{0}\|\alpha \right)=\Delta >0$. The ratio *R* of the statistical weight $W_{\beta }^{\beta }$ of all arrangements of $N_{\beta }$ individuals across all possible levels of $X_{0}$ resulting in the observed distribution $X_{0}|\beta$, and the statistical weight $W_{\beta }^{\alpha }$ of all arrangements of those $N_{\beta }$ individuals across all possible levels of $X_{0}$ resulting in a distribution, such as that observed for $X_{0}|\alpha$ is asymptotically given by,
(15)$$R=\frac{W_{\beta }^{\beta }}{W_{\beta }^{\alpha }}∼\exp\left(-N_{\beta }H\left(X_{0}\right)\Delta \right)$$

For large $N_{\beta }$ this ratio is vanishingly small. If the processes determining values of $X_{0}$ in sub-population β allowed for outcomes corresponding to a distribution, such as the one observed for $X_{0}|\alpha$, the likelihood we would instead observe the distribution we observe for $X_{0}|\beta$ would be practically zero. Repeated observation of Δ>0 leads us reasonably to conclude that those processes simply do not permit sub-population β from reaching the same distribution of $X_{0}$ as sub-population α.

### 5.2. Piecewise Decompositions for Two Sets of Characteristics

The coefficients inside the sum in (Equation 14) can be expressed in relation to two sets of individual degrees of freedom or characteristics as above. Denoting those sets by $X_{e}$ and $X_{i}$ this may be formally expressed as,
(16)$$\begin{array}{c} a\left(X_{0}\|\chi_{k}\right)=a\left(X_{0}\|\chi_{e}\right)+I_{\chi_{i}|\chi_{e}}=a\left(X_{0}\|\chi_{i}\right)+I_{\chi_{e}|\chi_{i}}\\ =I_{\chi_{i}|\chi_{e}}+I_{\chi_{e}|\chi_{i}}+m\left(X_{0},\chi_{e},\chi_{i}\right) \end{array}$$
where $I_{m|l}$ and $m\left(X_{0},m,l\right)$ are part-pointwise versions of the coefficients of incremental association defined in (Equation 12). The relationship between the part-pointwise coefficient of mutual association and its population-wide version follows trivially,
(17)$$\begin{array}{c} M\left(X_{0},X_{e},X_{i}\right)=\sum_{\chi_{e},\chi_{i}}f\left(\chi_{e},\chi_{i}\right)m\left(X_{0},\chi_{e},\chi_{i}\right)\\ \mathrm{where},\\ m\left(X_{0},\chi_{e},\chi_{i}\right)=a\left(X_{0}\|\chi_{e}\right)-I_{\chi_{e}|\chi_{i}}=a\left(X_{0}\|\chi_{i}\right)-I_{\chi_{i}|\chi_{e}}\\ =a\left(X_{0}\|\chi_{e}\right)+a\left(X_{0}\|\chi_{i}\right)-a\left(X_{0}\|\chi_{k}\right) \end{array}$$

This coefficient reflects an important aspect of the informational association between $X_{0}$ and pairs $\chi_{k}=\left(\chi_{e},\chi_{i}\right)$ of individual characteristics. As with the population coefficient of mutual information, any pair with $m\left(X_{0},\chi_{e},\chi_{i}\right)<0$ can be understood to have a pointwise “synergistic” informational association with $X_{0}$. For such pairs, their incremental informational association coefficients, $I_{\chi_{e}|\chi_{i}},I_{\chi_{e}|\chi_{i}}$ are greater than their respective unconditional coefficients of informational association, $a\left(X_{0}\|\chi_{e}\right)$, $a\left(X_{0}\|\chi_{i}\right)$. Equivalently, their joint coefficient of informational association with $X_{0}$ is greater than the sum of their respective coefficients of informational association with $X_{0}$. There is information about individual income in the combination of characteristics $\left(\chi_{e},\chi_{i}\right)$ that is not contained in either of those characteristics by themselves. Conversely, characteristic pairs $\left(\chi_{e},\chi_{i}\right)$ for which $m\left(X_{0},\chi_{e},\chi_{i}\right)>0$ have measures of redundancy in their informational association with *Y*. Please note that pairs $\left(\chi_{e},\chi_{i}\right)$ can be redundant or synergistic even when the sets $X_{e}$ and $X_{i}$ are synergetic or redundant, respectively.

The piecewise mutual information coefficient has a more general interpretation and significance. It can also be understood as a negative comparative measure of informational association between a set of characteristics $\chi_{e}$ and $X_{0}$ for a subpopulation $\chi_{i}=\alpha$, relative to the overall informational association between $\chi_{e}$ and $X_{0}$ for the entire population: $m\left(X_{0},\chi_{e},\alpha \right)=a\left(X_{0}\|\chi_{e}\right)-I_{\chi_{e}|\alpha }$. It may thus be taken as a (negative) measure of the comparative informational association of characteristics $\chi_{e}$ and $X_{0}$ across different $\chi_{i}$ sub-populations.

## 6. Application to Income and Social Identity

Indices of joint, mutual, and incremental informational association can be widely applied across different fields of social inquiry. For instance, they suggest a new methodological approach to debates concerning the possible independent influence of capital-market prices on the levels of investment undertaken by corporations [45,46,47]. It may also offer new ways to approach important questions such as disparities in criminal sentencing across race [48], or estimation of the independent added-value of education at elite educational institutions that recruit from very specific populations [49].

Here we briefly illustrate the use, novelty, and analytical power of these indices with an application to a burning question of political economy: The persistent economic effects of discrimination by gender, race, ethnicity, and other elements of *social identity*.

A large literature has provided evidence of the negative effects different forms of discrimination have on the incomes of members of certain social-identity groups [50,51,52,53,54,55]. One of the difficulties in grappling with the economic content and measure of those effects stems from the sheer complexity of the interrelationships involved in the determination of individual levels of income. Observable labor-market outcomes reflect the dynamic accumulation of educational, personal, and professional outcomes along an individual’s life [56,57,58,59]. The mechanisms and interactions linking any given observable individual characteristic and income are manifold, path-dependent, and very often unobservable. This includes the influence of discriminatory treatment, biases, and stereotypes in conditioning outcomes at all stages of those processes [60,61,62,63,64,65,66,67,68,69].

The complexity of the relationships involved create unsurmountable problems for statistical tests for discrimination in economic outcomes based on estimation of linear regression models [1]. Those tests effectively consider a joint hypothesis: The presence of an independent influence of identity on income and the specification of the model of the determination of wage income being used. The accuracy of the postulated tests for the presence of the former effect hinges entirely on the validity of the latter. Practical difficulties of estimation due to the omission of variables and other specification errors, multicollinearity, endogeneity among regressors, etc. seriously limit the usefulness of those diagnostic tests.

Indices of informational association fare much better. They offer simple, non-parametric ways to characterize the reduced-form associations between economic characteristics, social identity, and income. Since social identity is generally prior to the interactions involved in the determination of income, its associations with other degrees of freedom can be understood as measures of its total, direct and indirect influence over them. We can estimate their measure even without knowledge of the particular mechanisms or the full set of observed and unobserved degrees of freedom involved.

Large-scale data on individual incomes and categorical measures of social identity, age group, and level of educational attainment gathered in the decennial U.S. Census allows estimation of those indices and measurement of these influences. Those estimates cast new light onto the very nature of economic discrimination.

Before turning to this evidence it is useful to situate the application of indices of informational association to the study of income distributions in relation to existing uses of entropy to characterize inequality in income distributions, and to debates concerning equality of opportunity in economic systems. In both counts the approach developed above enables original contributions.

### 6.1. Inequality Indices, Identity, and Equality of Opportunity

In considering indices of informational association between incomes and two sets of categorical variables, we are effectively comparing the entropy of income distributions across sub-populations defined by those variables. This reveals both the close relationship and the important difference between these indices, when applied to patterns of income, and the entropy-based index of income inequality proposed by Henri Theil [2].

Theil’s index is defined for a population with i=1,…,N members, each of which has a share in total income of $y_{i}$,
(18)$$T=\log N+\sum_{i=1}^{N}y_{i}\log y_{i}$$

This index is well known to possess the requisite properties of an inequality measure, including obeying the Pigou-Dalton principle and exhibiting sub-group decomposability.

The entropy measure in Equation (Equation 1) applied to distributions of income over any given sub-population is different from this index. It is defined as a sum not over individuals but over coarse-grained income levels. This does not result in a measure of income inequality. It does not generally follow the Pigou-Dalton principle, doing so only for income distributions that are monotonically decreasing on income, and does not exhibit sub-group decomposability.

However, it is a sound measure of heterogeneity or uncertainty [29]. Most importantly, differences in its value across sub-populations allow informational characterizations of associations between income and other quantities established by the functioning of the economic system in question. This enables inferences about the functioning of an economic system and the processes conditioning levels of individual income.

Measures of informational association between individual characteristics and income also offer an innovative empirical diagnostic for the presence of *equality of opportunities* across different groups in a decentralized market economy.

Social-identity characteristics are a distinctive type of “circumstantial” or “arbitrary” factor that according to contemporary proponents of “equality of opportunity” should not affect income distributions: They are logically and almost always temporally prior to the processes determining an individual’s economic characteristics and their income. Race and ethnicity groupings are social creations with no biological foundation. The genetic variability across the different sets of human populations that constitute various racial and ethnic categories are very small compared to the overall genetic variability across humanity as a whole [70,71]. Any observed differences in patterns of economic behavior and characteristics by those categories are *social constructions*.

Sex is obviously a biological category, and there may well be economic characteristics and behaviors that are irreducibly sex-dysmorphic, that is, not the product of social processes of conventions [66]. The lack of counterfactual evidence prevents serious investigation of this strong claim. However, the *economic* consequences of any inherent sex-dysmorphic economic behavior or characteristics are expressions of gender, and reflect how a society attaches significance and economic value to that behavior and any other characteristics deemed “feminine” [72]. There is no *a priori*, biological justification for allocating the social product in ways that systematically disadvantage more than half of any society.

If an element of social identity is informative, we can conclude that it is influencing the social processes shaping individual economic characteristics, and the processes establishing incomes on the bases of those characteristics. These influences ensure some groups enjoy a narrower range of effective opportunities and, thus, incomes than others. That narrowing is a form of discrimination, effected by socio-economic processes, conventions, and institutions that systematically treat individuals differently on the basis of their social identity.

### 6.2. Data and Observation

We considered four waves of the U.S. Census data, from 1970 to 2000, as well as the 2007–2011 pooled American Community Survey, extracted from [73]. These surveys provide the most comprehensive and nationally representative source of data for income estimates across various subpopulations in the United States. For each respondent reporting market income, we observe their annual wage *Y*, and a set of two categorical economic characteristics $\chi_{e}$: age and level of educational attainment. We construct age-group information by decade, and four distinct levels of educational attainment. We also observe two social-identity characteristics, gender and race/ethnicity. We considered the two reported genders and three race/ethnicity categories: white, black, and Hispanic. All of the association indices above were estimated across all subgroups for each of the four decennial samples. The Appendix A describes the construction of our sample in detail.

There is considerable variation in the distributions of income across sub-populations defined by $X_{e}$ and $X_{i}$. While a full account of those differences and their implications for our understanding of economic discrimination is provided elsewhere, [1] two persistent patterns in the data clearly stand out.

First, the informational associations between social identities and incomes within each educational-achievement group, formally given by $I_{\chi_{i}|\chi_{e}}$, exhibit a persistent and telling pattern. As shown in Figure 1, men persistently enjoy negative measures of informativeness for their gender, while women enjoy positive ones. Along similar lines, whites enjoy negative measures of informativeness for their race/ethnicity, while blacks and hispanics enjoy positive ones (the incremental information index for being white among college-educated individuals is only weakly negative in part because whites make up the overwhelming majority of college educated individuals—86 percent of the total in our sample in 2010. In this setting, the positive values for non-white sub-populations capture the difference we are motivating almost entirely).

This implies we actually *gain* uncertainty about individual income when we learn that somebody is white or male. As groups, whites and men enjoy greater opportunities for income differentiation by characteristics *other* than their identity and observed level of educational attainment than everybody else. Women, blacks, and hispanics do not enjoy the same opportunities for potentially meritocratic differentiation. Their identities are associated with a wide variety of unobservable processes that effectively concentrate their realized market incomes, around values that are known to be lower than those enjoyed by men and whites. As shown in Figure 2, a similar pattern is observed across age groups.

These observed patterns are features of systems of discrimination that are irreducible to any individual mechanism, agency, or relationship. Yet they distinctively show how distributions of realized individual market incomes embody the very essence of pre-*judice*: The comparatively stronger extent to which some individuals are in effect treated “by the color of their skin,” (as well as gender, ethnicity, immigration status, etc.) instead of “by the content of their character” (as put by Martin Luther King, Jr. in his *I Have a Dream* speech of 28 August 1963).

The second feature involves estimated indices of piecewise mutual information $m\left(X_{0},\chi_{e},\chi_{i}\right)=a\left(X_{0}\|\chi_{e}\right)-I_{\chi_{e}|\chi_{i}}$. As discussed above, those piecewise indices can be understood as comparative measures across social-identity groups of the informational association a set of economic characteristics $\chi_{e}$ has with income. Their values for each sub-population $\left(\chi_{e},\chi_{i}\right)$ are shown in Figure 3, which relates them to the average income ${\bar{Y}}_{e,i}$ for that group measured relative to the average income across all observed individuals, $\bar{\bar{Y}}$. We denote that relative measure of average group income by $S_{e,i}$.

The figure conveys several well-understood features, iniquities, and developments in the distribution of income in the U.S. over the past four decades. One set of related features is particularly striking.

The curves for all groups of women have a distinctive “tilt,” ensuring they are almost always upward sloping in the plotted space. They exhibit lower measures of $m\left(Y,\chi_{e},\chi_{i}\right)$ for low levels of education, which correspond to low average levels of pay. There is a comparatively strong relative incremental informational association between their low levels of education and their incomes. In contrast, all groups of women have higher measures for $m\left(Y,\chi_{e},\chi_{i}\right)$ for high levels of education and pay. They enjoy comparatively weak incremental informational associations between their high levels of education and their incomes. Put differently, their educational level becomes a comparatively weaker informational predictor of their incomes as the level of education rises. It is a stronger informational predictor as it falls. In contrast, the curves for white men have a clear negative “tilt.” As their educational attainment levels increase, educational achievement becomes a comparatively *stronger* informational predictor of their incomes.

At the broadest level, these observations strongly suggest that social patterns of discrimination ensure the returns on educational achievement are very unevenly distributed across social-identity groups. Members of some groups are effectively punished more harshly than others for their low levels of education, while members of other groups are effectively rewarded more generously for their high levels of education. This has important implications for our understanding of the possibilities and limits of individual and social interventions seeking to reduce the economic consequences of discrimination.

## 7. Conclusions

This paper was motivated by a critical appreciation of the conceptual and practical problems faced by observational work founded on the strong form of individualist reductionism dominating current economic thought. The ordinary and piecewise informational indices it developed offer an innovative way to guide robust observational inquiry into reduced-form, systemic associations between a variable of interest and two sets of covariates in economic and broader social systems.

Their application to U.S. data on income, economic, and social-identity characteristics yields robust, observationally grounded insights into the economic effects and challenges posed by systems of discrimination in that economy. It also illustrated how information theory offers a natural language to express and investigate realities of social discrimination, which ensure elements of social identity are informative of observable economic outcomes.

We believe further work developing and applying indices of information association can make an important instrumental contribution to the development of observationally grounded insights into the functioning of complex economic and broader social systems.

## Figures and Tables

**Figure 1 entropy-21-00367-f001:**
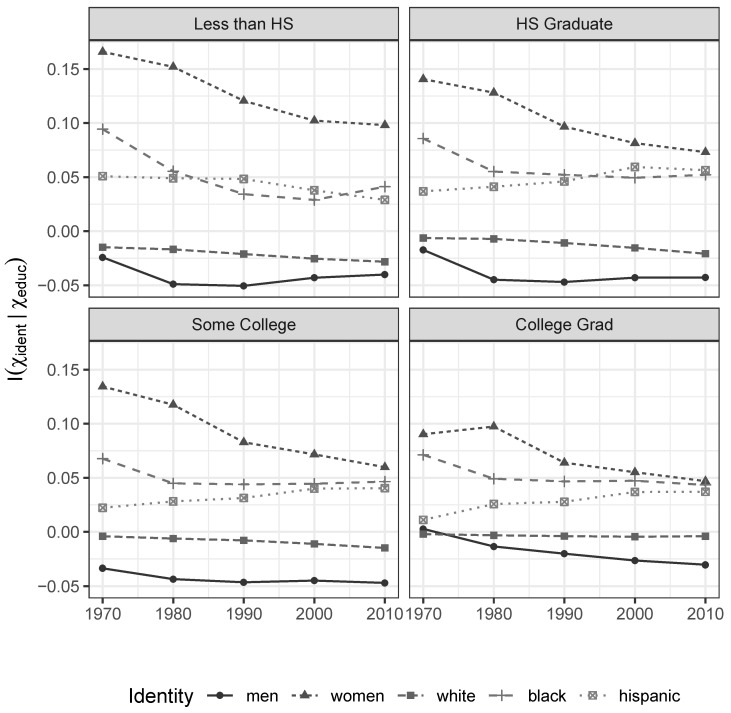
Incremental Informational Association of Gender and Race/Ethnicity Characteristics for given Levels of Educational Attainment. 1970–2010 Census and ACS data.

**Figure 2 entropy-21-00367-f002:**
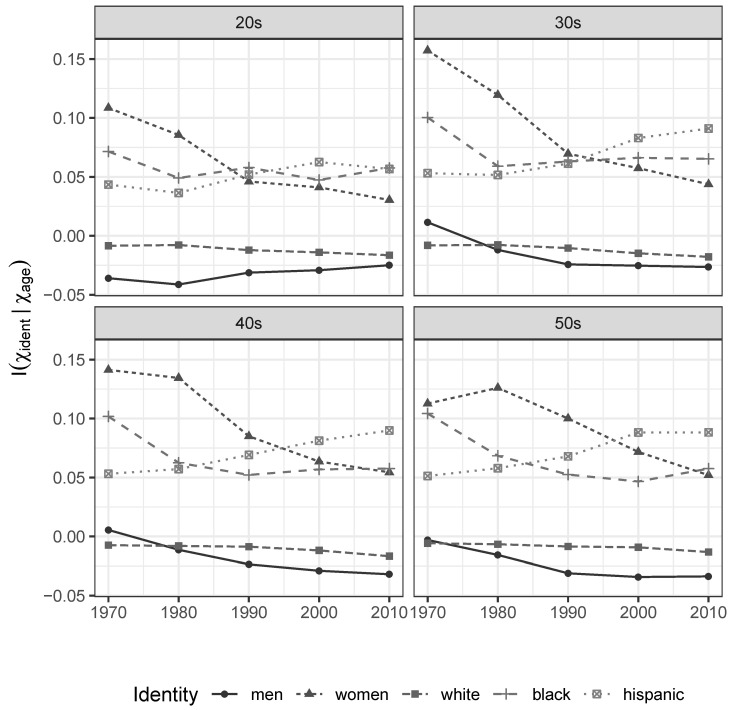
Incremental Informational Association of Gender and Race/Ethnicity Characteristics, given Age. 1970–2010 Census and ACS data.

**Figure 3 entropy-21-00367-f003:**
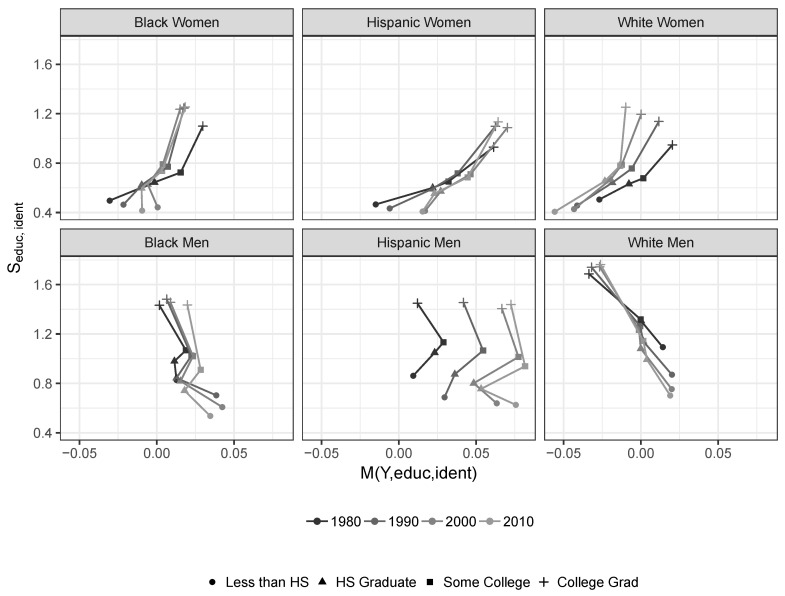
Mutual information between income, education, and identity, and relative average income of education level in question. 1980–2010 Census and ACS data.

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
