# Peer review of "Indices of Informational Association and Analysis of Complex Socio-Economic Systems"

_entropy, 2019, doi:10.3390/e21040367_

Round 1
Reviewer 1 Report
This manuscript develops new indices to tackle with the complex relationship among minority discriminated groups and economic measures, such as income, education and others. The authors have been empowered to pursue such task, and I am glad to see that they have crossed a long journey to reach a well-written, organized and very inspiri piece of science.
However, I would like to raise some questions that needed to be answered before its publication.
1) Although, the authors have brought some discussion on discrimination, there is a plenty of work done by authors such as Allport, Fisk and Guinote, with a lot of data that could shed some light in such topics. I would strongly advise to see works done by Guinote on Social power.
2) Although, the illustration is given in a complex system, as US economy, a second or third example would be important to show the cross-cultural features and applicability of such method. Otherwise, the authors should narrow the scope of the paper, starting from the title, for a one-case study.
Cuddy, A. J., Fiske, S. T., Kwan, V. S., Glick, P., Demoulin, S., Leyens, J. P., ... & Htun, T. T. (2009). Stereotype content model across cultures: Towards universal similarities and some differences. British Journal of Social Psychology, 48(1), 1-33.
Durante, F., Capozza, D., & Fiske, S. T. (2010). The stereotype content model: The role played by competence in inferring group status. TPM. Testing, Psychometrics, Methodology in Applied Psychology, 17(4), 187.
3) Also, the authors should compare their quantitative results with similar in the literature.
4) Figures should be better described. The legends should give a comprehensive description of the image. The quality of figures should also be improved.
5) The references should be updated. Apart from own publication there are few papers in 2018 and 2017.
6) There are also occasional misprints and wrong grammar in the main text. The authors should polish this before re-submission.
- p. 2: the the => the
- p. 23: to posses => to possess
- p. 24: persistenly => persistently
- p. 30: liminations => limitations
Author Response
Please see the attached .pdf file

Reviewer 2 Report
Summary:
This paper introduces and demonstrates the use of two indices of informational association. These indices indicate to researchers the level of information gained by the inclusion of a variable, and the potential incompleteness of the model to predict a given dependent variable. The authors illustrate with an application case the use of their indices.
Context and relevance:
The paper continues a narrow but relevant discussion in quantitative analysis, especially in the field of economics. To a limited audience, the paper has an interesting potential. The authors seem torn between introducing a discussion about social inequalities and systemic discrimination, and the presentation of the construction of the indices. It gives the impression that the indices were created to solve a very specific application case, while it seems it could be used more generally. I would suggest either embracing this and adjust the contextualization in the introduction and background analysis, or open the scope and give more concrete application examples. Probably the first option would be the most realistic.
A more vulgarized explanation of the usefulness of the indices given early in the paper would be beneficial, for example, right at the beginning rather than starting the introduction with the description of the paper.
Theoretical background:
Sections 2 and 3 are well structured and well written. I would say the main issue to these sections is the lack of a proper contextualization in the introduction, which forces the reader to go back to the abstract to be reminded exactly what the paper is trying to achieve.
Application case:
Comprehensive and understandable even for a reader less familiar with the domain.
Conclusion:
As a niche paper, it’s accessibility and potential audience is limited, but nonetheless interesting and, for the most part, well-written and well-argued. Its main flaw is an introduction that does not properly introduces the context and relevance. A suggestion for improvement would be to embrace the narrower application field of gaining information about the completeness of explanatory models in socio-economic inquiries about subgroup discrimination instead of general economics, and possibly extend the usefulness in the conclusion.
Round 2
Reviewer 1 Report
Dear authors,
I believe that all my questions have been properly answered.